# Mapping the quality of Norwegian health information –Does it facilitate informed choices?

Jürgen Kasper[1]*, Betül Cokluk[1], Marianne Molin[1,2], Anke Steckelberg[3], Sandro Zacher[3], the MAPPinfo project group[1,4¶], Victoria T. Hjellset[1]

1 Department of Nursing and Health Promotion, Faculty of Health Sciences, Oslo Metropolitan University, Pilestredet, Oslo, Norway, 2 Department of Health and Exercise, School of Health Sciences, Kristiania University College of Applied Sciences, Kirkegata, Oslo, Norway, 3 Institute of Health, Midwifery and Nursing Science, Medical Faculty of Martin Luther University Halle-Wittenberg, University Medicine Halle, Halle (Saale), Germany, 4 Department of Health and Caring Sciences, Faculty of Health Sciences, UiT, The Arctic University of Norway, Langnes, Norway

¶ Membership of the MAPPinfo project group is provided in the Acknowledgments.
* jurgenka@oslomet.no

## Abstract

### Background and aim

Health literacy refers to the ability to use relevant information to make informed choices. However, the quality of the available information influences how well individuals can make those choices. Evidence-based recommendations for the development and design of health information have recently been published. In this study, we aimed to map the quality of Norwegian web-based health information across selected public health domains.

### Methods

Using a multiple-cross-sectional design, we assessed information in 16 health domains relevant to infants, children, and youth. Convenience samples were drawn using structured Google searches. Three independent raters conducted the quality appraisal by applying the 19 criteria of the *Mapping the quality of health information* checklist. Inter-rater reliability was calculated using T-coefficients. Information quality was statistically described. To explain variance in quality, mean quality scores were compared across three independent variables: the type of the health problem, target group, and provider class.

### Results

Across the surveys, 1,948 health information materials from 64 subdomains were assessed. Inter-rater reliability was excellent (mean T = .89/.90). On average, the materials complied with 22% (range: 0–73%, standard deviation = .09) of the current minimal standard. Differences between types of problems or target groups were

**Data availability statement:** All relevant data are within the manuscript, a supporting EXCEL file naming all evaluated websites and published on "Zenodo" (A reference to the data file is provided in the manuscript).

**Funding:** The study was conducted as part of the project "Quality of health information: the missing link in the era of health literacy" which is funded as a three years PHD scholarship by the faculty of health sciences, OsloMet, Metropolitan University, Oslo, Norway (Project number: 203322). The author, Betül Cokluk, is the PhD fellow funded by this project.

**Competing interests:** The authors have declared that no competing interests exist.

marginal. No differences were found between information provided by health authorities, health services, or commercial entities.

## Conclusion

Norwegian web-based health information is not of sufficient quality to facilitate informed health choices made by citizens. These findings apply across a wide range of public health domains relating to infants, children, and youth. In the absence of appropriate health information of acceptable quality, estimates of the public's level of health literacy may need reconsideration. Further research is needed to appraise the quality of information in other health domains and countries.

## Background

The age of evidence-based practice has redefined the responsibility for making health-related decisions. Previously, specialists such as doctors, as well as health authorities, were expected to identify and recommend the correct course of action among several possible alternatives. However, ordinary citizens–the lay public— are now encouraged to make their own health decisions, either in cooperation with health care professionals or independently, depending on the type of decision. This approach can support good health outcomes only when good-quality information is available, enabling individuals to be well informed about their choices [1,2]. Informed choices are based on relevant knowledge, consistent with the decision-maker's values, and behaviourally implemented [2].

Reflecting a consensus within scientific communities and many societies worldwide, the concept of "informed choice" has become an important quality parameter of health care provision. It is embedded in international ethical standards [3,4], national patient rights [5], and numerous health professional guidelines [6]. This implies that health service users, such as patients in hospitals, visitors to municipal health care centres, or national health information platform users, expect support in making informed decisions about their health. Such support includes forms of communication that facilitate understanding and the intellectual processing of information relevant to the pending decision [7,8].

Many, health-related decisions are made by individuals without involving health care providers and, therefore, without these individuals necessarily becoming formal health service users. Examples include decisions about whether to visit a doctor for a health concern, whether to cycle or drive to work, or whether to use hormonal contraception. Most people seek health-related information through Internet search engines and do not base their decisions solely on information provided by health professionals [9].

The way people search for, select, and use information varies between individuals and is guided more by intuitive than by rational processes. These intuitive processes are shaped by emotional states, social contexts, economic conditions, motivations, preferences, literacy, and various intellectual abilities [10]. The unique nature of such

intuitive processes creates opportunities for companies such as Google or Meta to systematically influence decision-making behaviour by analysing individual search patterns and altering personal search algorithms.

The quality of health choices is also influenced by individual competencies to evaluate the quality and reliability of information and to avoid being influenced by outside forces, in order to make choices that lead to good health outcomes—a concept subsumed under the term health literacy [11]. A recent review showed that health literacy is low in the Norwegian population [12]. In response, the Norwegian government has developed a strategy to strengthen health literacy across its population [13]. Amongst other anticipated benefits, Norwegians may make better use of existing health information and adopt healthier lifestyles. However, for this to occur, at least some high-quality information sources must be available.

Information that supports informed decision-making is called evidence-based health information (EBHI) [14]. "Evidence-based" refers to both the content and presentation of information. Research over several decades has examined how to develop and design health information that facilitates informed health choices. This body of evidence is summarised in the guideline "Evidence-based health information" [15], which provides updated evidence on 21 research questions relating to the development and design of health information. In addition, the guidelines summarises previously published ethical guidelines [4]. Specifically, the guideline's recommendations concern transparency regarding the background of the authors and the origin of the content, the completeness of the content, its presentation considering known sources of bias or misunderstanding, the methods used to search for, select, and evaluate information during development, and, finally, measures to involve the target group in developing the information material and documentation of the evaluation of suitability for the target group [15].

Having in mind the current emphasis on health literacy and its definition as a competency to process information, we believe that the most important question is whether citizens in our country have suitable information available that they can process. Our project focuses on information about health problems that do not necessarily require the involvement of health care providers and that are relevant to the target group of Public Health Nurses (PHN). Norwegian public health nursing is a unique service addressing children and youth (aged 0–20 years) and their families using a health-promoting and preventive approach. Working at the municipal level in schools and health care centres, PHNs play a crucial role in educating and empowering children, youth, and families so that they can make well-informed health decisions. For many health issues relevant to these groups, the function of PHNs involves being accountable for counselling and information provision when needed, although they do not necessarily have a formal role in making the associated health decisions.

Robust evidence already exists regarding the evaluation of health information provision, using various concepts of quality and diverse evaluation methods [16–23]. In a previous study, we searched for evaluation methods that complied with evidence-based quality criteria but did not find any [24]. We therefore assume that no systematic evaluation of health information quality has been conducted based on the current guidelines [15]. Moreover, to the best of our knowledge, this type of research has not been conducted in, where we are affiliated.

In this study, we aimed to map the quality of Norwegian web-based health information materials (WBHIMs) in selected public health domains relevant to the clientele of Norwegian PHNs. Specifically, we aimed to determine whether Norwegian citizens can make informed health choices based on the information available on the Internet (research question 1). We also aimed to provide insight into the nature of any potential shortcomings (research question 2). Moreover, three independent variables—the type of the health problem, provider class, and target group—were tested for their potential contribution to explaining variance in the quality of Norwegian health information (research question3).

## Methods

### Design

In this study, we used a multiple cross-sectional design to map the quality of openly accessible health information on the internet in selected health domains relevant to the clientele of Norwegian public health nursing services. The term 'health

domain' refers to superordinate topics, such as sleeping problems in infants or perimenstrual complaints. Within the research design, scrutiny was focused at the level of subdomains, such as prevention of sudden infant death, treatment of delayed sleep onset, or management of menstrual pain or heavy bleeding. Each subdomain was defined analogously to a medical indication by: first, a specific diagnosis; second, the type of measures (diagnostic, treatment, prevention, health promotion, or rehabilitation); and third, a particular target group. Each cross-sectional study analysed the quality of all identifiable WBHIMs within a specific subdomain Fig 1.

The current analysis integrates the findings of 64 cross-sectional studies across 16 health domains (Table 1). All the studies were conducted as part of the MAPPinfo (Mapping quality of health information) project and carried out by 21 master's students in two separate cohorts, 1 year apart, using the same method. We designed this study from the user's perspective given that it is the individual who seeks out health information in the majority of cases and thus to whom we want to generalize our results. Adopting this perspective guided decisions about which health domains and subdomains to examine. We also informed the development of the search strategy used to identify websites, based on how we assumed a typical user would proceed. Finally, the user's perspective is reflected in the choice of the quality concept [15], which considers informed health choices as the crucial target [2].

The Strengthening the Reporting of Observational Studies in Epidemiology guideline (STROBE [25]) was applied in reporting the current study.

## Sample

Considering that it would not be possible, necessary, or meaningful to appraise all WBHIMs within the chosen sector of the Norwegian health information landscape, we established a system for selecting health domains and specific subdomains from the broader map. Choices about which health domains to screen were designed to avoid arbitrariness and to reflect the users' perceptions of relevance.

In the first cohort of 28 surveys conducted by master's students, we selected domains related to infant health under 12 months of age. To determine which themes might be considered relevant by the target group, we conducted a pre-study.

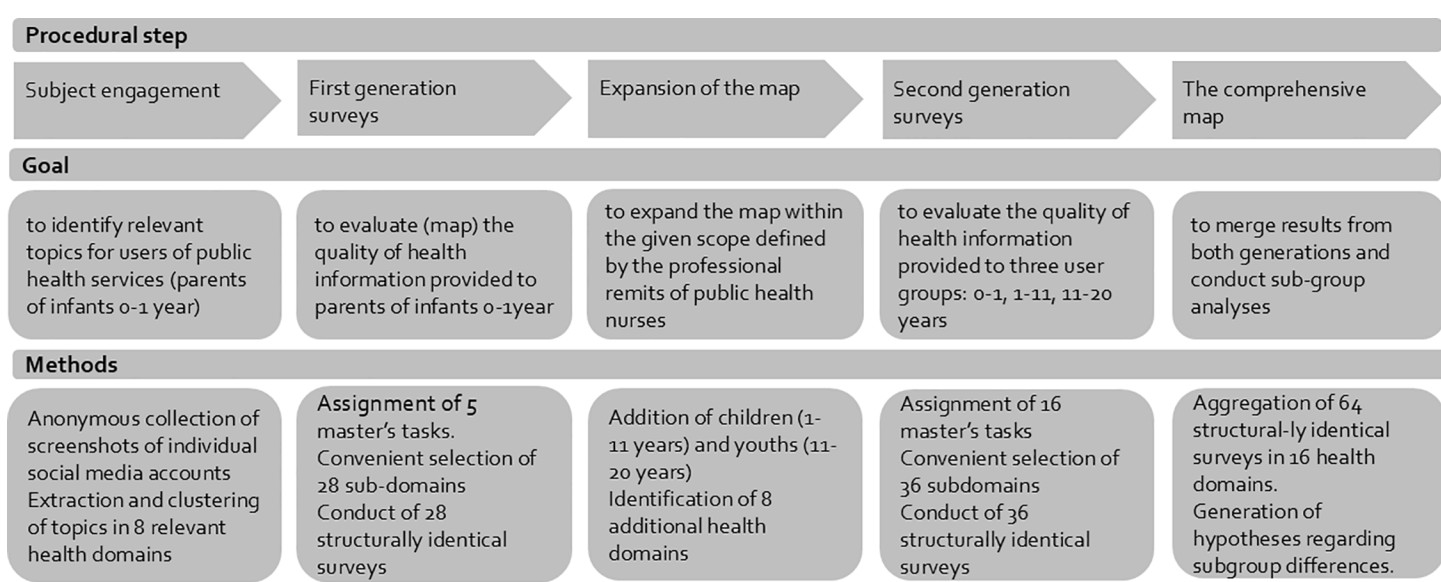

**Fig 1. Overview of the multiple cross-sectional study procedure.** This figure illustrates the procedural steps followed during the conduct of 64 structurally identical surveys.

**Table 1. Description of the sample of WBHIMs and results on information quality.**

| Health domain | | Number | Decision type | Target group | Information quality (%) | | |
|---|---|---|---|---|---|---|---|
| | | | | Subdomain | Mean | Min/max | SD |
| 1 Breastfeeding | | 59 | | | .24 | .05/.45 | .08 |
| | Mastitis[1] | 26 | treat | Infants | .24 | .15/.40 | .07 |
| | Sudden termination[1] | 8 | treat | Infants | .26 | .18/.38 | .07 |
| | Type of nutrition[1] | 25 | prom | Infants | .27 | .05/.45 | .09 |
| 2 Eating disorder Anorexia | | 74 | treat | Youth | .20 | .08/.50 | .08 |
| 3 Infant nutrition | | 108 | | | .23 | .05/.68 | .08 |
| | Eating difficulties[1] | 11 | prev | Infants | .18 | .05/.28 | .08 |
| | Food allergy[1] | 11 | prev | Infants | .33 | .18/.68 | .15 |
| | Iron deficiency[1] | 29 | prev | Infants | .24 | .10/.35 | .06 |
| | Vegan diet[1] | 11 | prev | Infants | .19 | .13/.33 | .05 |
| | Vegetarian diet[1] | 12 | prev | Infants | .16 | .08/.33 | .07 |
| | Vitamin D deficiency[1] | 34 | prev | Infants | .23 | .15/.33 | .05 |
| 4 Menstruation complaints | | 99 | | | .21 | .03/.45 | .09 |
| | Pain | 39 | treat | Youth | .20 | .05/.40 | .09 |
| | Heavy bleeding | 19 | treat | Youth | .19 | .03/.40 | .11 |
| | Irregular menstr. | 20 | treat | Youth | .23 | .10/.45 | .09 |
| | Missed menstr. | 21 | treat | Youth | .24 | .13/.40 | .06 |
| 5 Mental health | | 217 | | | .22 | .00/.50 | .09 |
| | OCD | 42 | treat | Youth | .19 | .00/.05 | .08 |
| | PTSD | 26 | treat | Youth | .17 | .00/.30 | .07 |
| | Depression | 39 | treat | Youth | .21 | .08/.43 | .08 |
| | Social anxiety | 21 | treat | Youth | .22 | .10/.38 | .07 |
| | Attachment ability[1] | 47 | prom | Infant | .25 | .08/.43 | .09 |
| | Attachment disorder[1] | 42 | prev | Infant | .26 | .08/.43 | .08 |
| 6 Overweight | | 182 | | | .19 | .00/ 43 | .09 |
| | Overweight children | 121 | treat | Children | .19 | .00/.43 | .09 |
| | Overweight youth | 61 | treat | Youth | .19 | .05/.43 | .07 |
| 7 Pregnancy | | 103 | | | .18 | .05/.33 | .06 |
| | Contraception | 52 | prev | Youth | .17 | .05/.33 | .06 |
| | Unintended pregn. | 51 | treat | Youth | .19 | .05/.33 | .05 |
| 8 Psychoactive subst. misuse | | 302 | | | .22 | .05/.45 | .08 |
| | Addiction alcohol | 45 | prev | Youth | .25 | .08/.45 | .07 |
| | Addiction cannabis | 30 | prev | Youth | .21 | .05/.35 | .08 |
| | Addiction cocaine | 26 | prev | Youth | .17 | .10/.38 | .08 |
| | Immed. harms alcohol | 43 | prev | Youth | .24 | .15/.38 | .06 |
| | Immed harms cannabis | 34 | prev | Youth | .21 | .05/.35 | .08 |
| | Imme.harms cocaine | 27 | prev | Youth | .22 | .10/.38 | .08 |
| | Long-term harms alcohol | 40 | prev | Youth | .21 | .05/.40 | .07 |
| | Long-term harms cannab. | 30 | prev | Youth | .21 | .05/.35 | .08 |
| | Long-term harms cocaine | 27 | prev | Youth | .22 | .10/.38 | .08 |
| 9 Screen misuse | | 81 | | | .20 | .05/.50 | .10 |
| | Cognitive development | 28 | prev | Children | .22 | .10/.50 | .11 |
| | Overweight | 18 | prev | Children | .20 | .10/.36 | .08 |
| | Sleeping problems | 35 | prev | Children | .19 | .05/.40 | .09 |
| 10 Sexually transm. diseases | | 52 | | | .17 | .05/.35 | .07 |

*(Continued)*

**Table 1.** (Continued)

| Health domain | | Number | Decision type | Target group | Information quality (%) | | |
|---|---|---|---|---|---|---|---|
| | | | | Subdomain | Mean | Min/max | SD |
| | Chlamydia | 30 | prev | Youth | .17 | .05/.35 | .07 |
| | Gonorrhea | 22 | prev | Youth | .16 | .05/.35 | .08 |
| 11 Sleeping disorders | | 61 | | | .18 | .05/.38 | .08 |
| | Crib death[1] | 36 | prev | Infants | .18 | .05/.38 | .08 |
| | Sleeping difficulties[1] | 25 | treat | Infants | .18 | .05/.33 | .07 |
| 12 Subst. misuse breastfeeding | | 80 | | | .23 | .05/.57 | .10 |
| | Alcohol | 23 | treat | Infants | .28 | .10/.57 | .14 |
| | Caffeine | 26 | treat | Infants | .22 | .10/.40 | .07 |
| | Smoke | 19 | treat | Infants | .22 | .05/.40 | .08 |
| | Snuff | 12 | treat | Infants | .20 | .05/.30 | .09 |
| 13 Teething troubles | | 202 | | | .25 | .08/.73 | .09 |
| | Chickenpox[1] | 32 | treat | Children | .26 | .10/.43 | .07 |
| | Colics[1] | 43 | treat | Children | .29 | .10/.73 | .11 |
| | Otitis[1] | 42 | treat | Children | .25 | .13/.53 | .10 |
| | Respiratory infection[1] | 22 | treat | Children | .23 | .13/.35 | .06 |
| | Stomach bug[1] | 41 | treat | Children | .21 | .08/.40 | .08 |
| | Urinary tract infection[1] | 22 | treat | Children | .23 | .15/.40 | .06 |
| 14 Vaccine against HPV | | 116 | | | .20 | .05/.38 | .08 |
| | HPV boys | 54 | prev | Youth | .20 | .05/.38 | .08 |
| | HPV girls | 62 | prev | Youth | .20 | .05/.38 | .08 |
| 15 Vaccines infant | | 146 | | | .26 | .13/.43 | .07 |
| | Diphtheria[1] | 17 | prev | Children | .28 | .18/.43 | .08 |
| | Haemophilia[1] | 17 | prev | Children | .29 | .18/.40 | .07 |
| | Hepatitis B[1] | 14 | prev | Children | .24 | .15/.38 | .06 |
| | Pneumonia[1] | 14 | prev | Children | .27 | .18/.40 | .07 |
| | Polio[1] | 19 | prev | Children | .25 | .13/.40 | .07 |
| | Rotavirus[1] | 15 | prev | Children | .26 | .18/.40 | .07 |
| | Tetanus[1] | 18 | prev | Children | .25 | .15/.40 | .07 |
| | Tuberculosis[1] | 13 | prev | Children | .27 | .18/.40 | .07 |
| | Whooping cough[1] | 19 | prev | Children | .26 | .18/.43 | .06 |
| 16 Vaccines children | | 66 | | | .24 | .05/.40 | .09 |
| | Measles | 31 | prev | Infants | .22 | .05/.40 | .09 |
| | Mumps | 17 | prev | Infants | .25 | .10/.40 | .08 |
| | Rubella | 18 | prev | Infants | .25 | .15/.40 | .08 |
| Total | | 1948 | | | .22 | .00/ 73 | .09 |

Description: prev = information about a problem related to prevention of a specific health state or outcome, treat = information about a problem related to treatment of a specific health state or disease, prom = information about how to promote a specific health state or outcome, WBHIM = web-based health information material. 1)=survey was conducted by the first cohort of masters' students.

In that pre-study, a convenience sample of twenty parents with children under 12 months of age was contacted via the students' personal networks and local health care centres. The parents were asked to collect screenshots of health information or health-related claims they encountered on social media over a period of 5 days (January 2023). Participants received a study information sheet, including instructions and a link to an online form. By sharing their screenshots

anonymously online, the parent-participants provided implicit informed consent for the use of their data in the research. This information was kept confidential, and the authors had no access to any data that could identify individual participants. The pre-study was approved by the Norwegian agency for Shared Services in Education and Research (SIKT) [26], reference number 768954.

Through this pre-study, we collected 228 screenshots after removing duplicates. The research group analysed the pool of screenshots and clustered them according to health domains within the given age range. The choice of subdomains to be mapped by the first cohort of studies was informed by this material, as students selected from the pool and specified and customised their individual theses. In the second cohort of 36 additional surveys, we extended target topics to include health domains relevant to children aged 1–11 years and youth aged 12–20 years. Identification of relevant domains for children's and youth's health was achieved partly by extrapolating from similar infant domains, such as vaccination or consumption of psychotropic substances, and partly by identifying additional domains with specific relevance to these age ranges, such as pregnancy.

Sampling in the current multi cross-sectional study refers to 64 populations of WBHIMs, corresponding to the 64 subdomains screened. Across all 64 cross-sectional studies, our recruitment strategy was designed to identify as many WBHIMs as a typical user might encounter in a Google search.

After clearing the cache on their browsers, the students drew convenience samples of WBHIMs using the following selection criteria: Norwegian language, non-professional content, and no barriers to access. WBHIMs were identified via structured Google searches, which varied across the 64 surveys but were developed following a common procedure. First, orienting searches were conducted using combinations of search terms likely to be chosen by a user. Results were compared with a test set of WBHIMs to assess redundancy between terms and to identify the core terms for the final search. Second, the final search was run, which included a cutoff criterion specifying how many references generated by the Google search should be considered.

## Measurement methods

Quality appraisal was conducted using the MAPPinfo instrument (Mapping the quality of health information checklist [27]), which is based on the evidence-based health information (EBHI) guideline [15]. The instrument is novel in three regards: First, it is the first to precisely operationalise the EBHI guideline recommendations [15,24]. Second, appraisal is based solely on the published information material, without requiring consultation of secondary sources. Third, the instrument can be used reliably by individuals without special training or a background in evidence-based practice, making quality appraisal more transparent and accessible to everyone.

Following a "pars pro toto" approach, MAPPinfo evaluates only a selection of criteria provided in the guideline. In particular, it includes criteria based on strong recommendations and excludes criteria that cannot easily be observed in health information documents or websites. Criteria relating to methods of development or the state of evaluation of a WBHIM are not included, as they cannot be assessed solely from the WBHIMs. MAPPinfo has been validated as a valid screening instrument and provides a very good estimate of the overall quality of information with reference to the EBHI guideline recommendations [27].

MAPPinfo is designed as a checklist comprising 19 criteria, which are thoroughly described and defined in a manual that also includes good practice examples. The criteria are organised into four categories: definitions (how the target group, topic, and goal of the information are identified and described); transparency (how accurately and clearly the authors, producers, funding sources, conflicts of interest, and origin of information are presented); content (how comprehensive the information is in relation to prevalence, natural course, benefits and harms of all alternatives, test reliability in the case of diagnostic options, and uncertainty); and presentation (whether the methods used to provide information are evidence-based, particularly methods for communicating quantitative information, such as the effects of treatments).

Some checklist criteria require consideration of several elements; however, the answering format is dichotomous or trichotomous, leaving little room to grade the quality of information. This rigorous method of appraisal was chosen by the developers of MAPPinfo, taking into account the user's need to see all elements of a criterion fulfilled in order to use the information effectively. For example, if the benefit is explained according to the guidelines in terms of absolute risk reduction for only one of three available options, the respective rating would be zero, because information on benefit is considered insufficient for a citizen until they can compare all options. A graded scale might better reflect the provider's skill level; however, from the citizen's perspective, the question is whether an informed choice is possible. MAPPinfo captures the essential elements of health information quality and therefore provides a minimum standard for evaluation. Other potentially relevant criteria may exist but are not yet known. In other words, partial compliance is never sufficient, while full compliance does not necessarily ensure high-quality health information.

### Data collection

All websites were classified according to three variables. The type of health problem was categorised as diagnostic, treatment, preventive, or health promotion. In accordance with the sampling structure, three target group clusters were defined: infants (0–12 months), children (1–11 years) and youth (12–20 years). Health information providers were classified as research, governmental, health service, non-governmental organisation (NGO), commercial entities, news organisations, or bloggers and influencers.

Quality appraisal of the WBHIMs was carried out via a rigorous process to ensure consistency in applying the criteria across the 64 surveys. This approach was appropriate for a research study, despite previous evidence that the checklist's psychometric properties are sufficient for use by untrained raters [27].

Ratings were documented in a specially prepared Microsoft Excel spreadsheet [28], which allowed data entry from different raters and recorded areas of disagreement. The spreadsheet generated result tables including reliability metrics, information quality, and a diagram providing a visual representation of the results. Before starting data collection, the master's students and supervising researchers (BC, VTH, JK) met to calibrate the application of criteria to specific health problems using examples of WBHIMs. Beyond strengthening inter-rater reliability, this meeting contributed to harmonising the application of the method across all 64 surveys.

Decisions were also made about which content would be considered part of a particular WBHIM versus content considered external. These decisions were generally based on the website URL. For example, an information website with hierarchically organised pages under a specific landing page from the same provider was treated as a single WBHIM, whereas a link to a neighbouring website under a different landing page or provided by another entity was not included as part of the same WBHIM. The WBHIMs were rated consecutively and independently by two master's students, and any disagreements were resolved through discussion. To provide a reference standard, the websites were also rated by experts in EBHI (BC, VTH, JK), and any disagreements between the students' consensus and the reference standard were resolved through discussion before a second consensus rating was documented. The first four of the 64 cross-sectional studies were conducted between February and April 2023 at the Arctic University of Norway in Tromsø and at OsloMet University in Oslo, and the remaining 60 studies were conducted between March and May 2024 at OsloMet.

### Analyses

Data were initially analysed at the subdomain level using MS EXCEL (version 2402) and subsequently transposed into SPSS (version 28.0.1.1) for broader-level analyses. Analyses were based on five data columns: columns one and two contained the individual ratings by the student raters, column three represented the agreement between the students, column four comprised the reference standard, and column five contained the final consensus.

**Reliability and validity.** To estimate the quality of the measurement process, inter-rater agreement was quantified. Regardless of the coefficient used, this calculation adjusts the number of observed matches for the number of matches

expected by chance. Inter-rater reliability between the students (columns 1 and 2) was calculated at the criterion level using the "T" coefficient [29], an adjusted Cohen's kappa [30]. Unlike the original kappa, this adjustment estimates the probability of expected matches theoretically, based on the potential values a criterion can take, rather than empirically from the data set. This adjustment reflects the assumption that raters should orient observations according to what is theoretically possible, unbiased from previous observations [31]. Inter-rater reliability coefficients range from 0 to 1 and were interpreted as moderate (0.40–0.60), strong (>0.60), or excellent (>0.80) [32]. Means of inter-rater reliability were calculated for each subdomain and rater teams.

The same statistics were used to compare the two student raters as a unit (column 3) with the reference standard (column 4) to estimate the validity of the preliminary results (first consensus). In the absence of a gold standard, the reference standard was considered suitable for calculating criterion validity. This was performed at the level of individual criteria and as mean validity scores. Due to additional expert appraisal and discussion after the first consensus, these validity scores are considered a conservative estimation of the validity of the final results (column 5).

**Quality of the information.** Each WBHIM was assigned a quality score (QS), representing a percentage (0–100) of criteria met. As each criterion operationalises a single evidence-based strong recommendation, the set of criteria constitutes a minimum standard. Since even a violation of one criterion can impede a fully informed choice, 100% of the criteria must be met to facilitate informed choices [27]. In addition, criteria scores were calculated to provide the percentage of WBHIMs within a subdomain that complied with each specific criterion. Quality scores are reported as means aggregated over all WBHIMs within specific subdomains, domains, and the total sample. Criteria scores are reported as means aggregated across the total sample of WBHIMs.

**Inference testing of subgroups.** To examine how information quality varies between materials targeting different groups, types of health problems, or published by different provider classes, three analyses of variance (ANOVAS) were performed. In all ANOVAs, the quality score was the dependent variable. In the first ANOVA, the independent variable was target group, with three levels (infants, children, and youths). In the second ANOVA, the independent variable was the type of health problem, with three factor steps (prevention, treatment, and health promotion). In the third ANOVA, the independent variable was provider class, with eight levels (research organisations, governmental entities, health service institutions, NGOs, commercial entities, news organisations, and bloggers/influencers). Differences within the sets of factor levels were examined using post hoc Scheffé tests for pairwise comparisons. P-values below 0.05 were considered statistically significant. Additional interpretation was guided by the corresponding eta-squared parameters, estimating the effect size of each p-value. This was particularly important as these inference tests were overpowered and the number of single tests was high, implying a risk of p-value inflation and random effects due to multiple testing.

## Results

The complete data file is accessible at https://zenodo.org/ [33].

## Description of the sample

Across the 64 cross-sectional studies in 16 health domains, we assessed a total of 1,948 WBHIMs presented on 1,538 websites (Table 1). Many websites were evaluated multiple times, focusing on different subdomains.

Commercial providers accounted for 45% (876) of the total sample, news and NGOs each accounted for 13% (256 and 259, respectively), public health services represented 10% (203), governmental entities 9% (184), scientific organisations 5% (103), and bloggers 3% (67).

## The quality of the measurement

Inter-rater agreement was excellent on average across the 64 subdomains (T = .89, standard deviation [SD]=.07) and for 59 of the 64 individual subdomains (T min = .73, T max = .99). Criterion validity was also excellent on average across the

64 subdomains (T = .90, SD = .06, T min = .57, T max = .99) and for 62 of the 64 individual subdomains. The sampling of information sites was relatively evenly distributed across the target populations (20 infants, 19 children, 25 youth). Subdomains addressing prevention were strongly represented (37 of 64 subdomains), treatment-related subdomains proportionally represented (25 of 64 subdomains), and information on health promotion was rare (two of 64 subdomains).

## The quality of the information

Results answering research question 1 about the level of information quality: Across the 1,948 WBHIMs evaluated, health information complied with an average of 22% of the quality criteria (SD = .09, min = 0%, max = 73%) (Table 1). This implies that WBHIMs disregard 78% of the minimal standards of quality on average. Table 1 provides detailed results for each investigated health domain and subdomain.

Results answering research question 2 about the nature of information quality within the given samples: These results focus on individual quality criteria and are detailed in Fig 2 and Table 2. Across the total sample, the percentage WBHIMs meeting individual criteria ranged from 0% to 82% (Table 2, Fig 2). The three most frequently met criteria were: "target group named" (55%), "language neutral" (72%), and "use of narratives avoided" (82%). Conversely, three criteria met by fewer than 1% of WBHIMs were: "systematic search strategy reported," "benefits and harms disclosed," and "gain-loss framing used."

## Moderators of information quality

This section reports results answering the third research question, which sought to explain variation in quality scores across the total sample. As shown in Table 1, the SD was low (SD = 0.09), indicating limited variation among the 1,948 quality scores around the mean value (0.22).

A significant main effect is observed for the factor target group; however, the effect size was small ($p < .001$, $\eta^2 = .05$). Post hoc pairwise comparisons showed that health information relating to infants (n = 745 of 1948) had slightly higher quality than information relating to children (n = 570), whereas information relating to youth (n = 633) had even lower quality than the other two groups ($QS_{infant} = 24\%$ min/max: 5/73; $QS_{children} = 21\%$ min/max: 0/50; $QS_{youth} = 20\%$, min/max 0/73; all pairwise comparisons significant: $p < .001$).

Similarly, a significant main effect was observed for decision type, although with a very small effect size ($p = .004$, $\eta^2 = .006$). Post hoc Scheffé tests indicated that information on health promotion was of higher quality (n = 72) than information on treatment (n = 875, $p = .006$) or prevention (n = 1001, $p = .032$) ($QS_{health\ promotion} = 25\%$ min/max:5/45; $QS_{prevention} = 22\%$, min/max: 5/68; $QS_{treatment} = 21\%$ min/max: 0/73).

Finally, a significant main effect was observed for provider class, with a small effect size ($p < .001$, $\eta^2 = .05$) (absolute numbers are provided in Table 3). Post hoc Scheffé tests show that information from scientific organisations had higher quality scores than all other six provider classes ($p < .001$). Information from NGOs had higher scores than information from news organisations ($p < .001$); however, it was not superior to any of the other five provider classes. Governmental information had slightly higher quality scores than information from news organisations ($p = .012$); however, this was not superior to commercial entities, health services, NGOs, or bloggers/influencers. No differences were observed between information from health service entities and that from commercial entities, news organisations, or bloggers/influencers.

# Discussion

## Short summary of aim and approach

In this study, we aimed to investigate whether openly accessible web-based information in Norwegian language is of sufficient quality to enable citizens to make informed health choices.

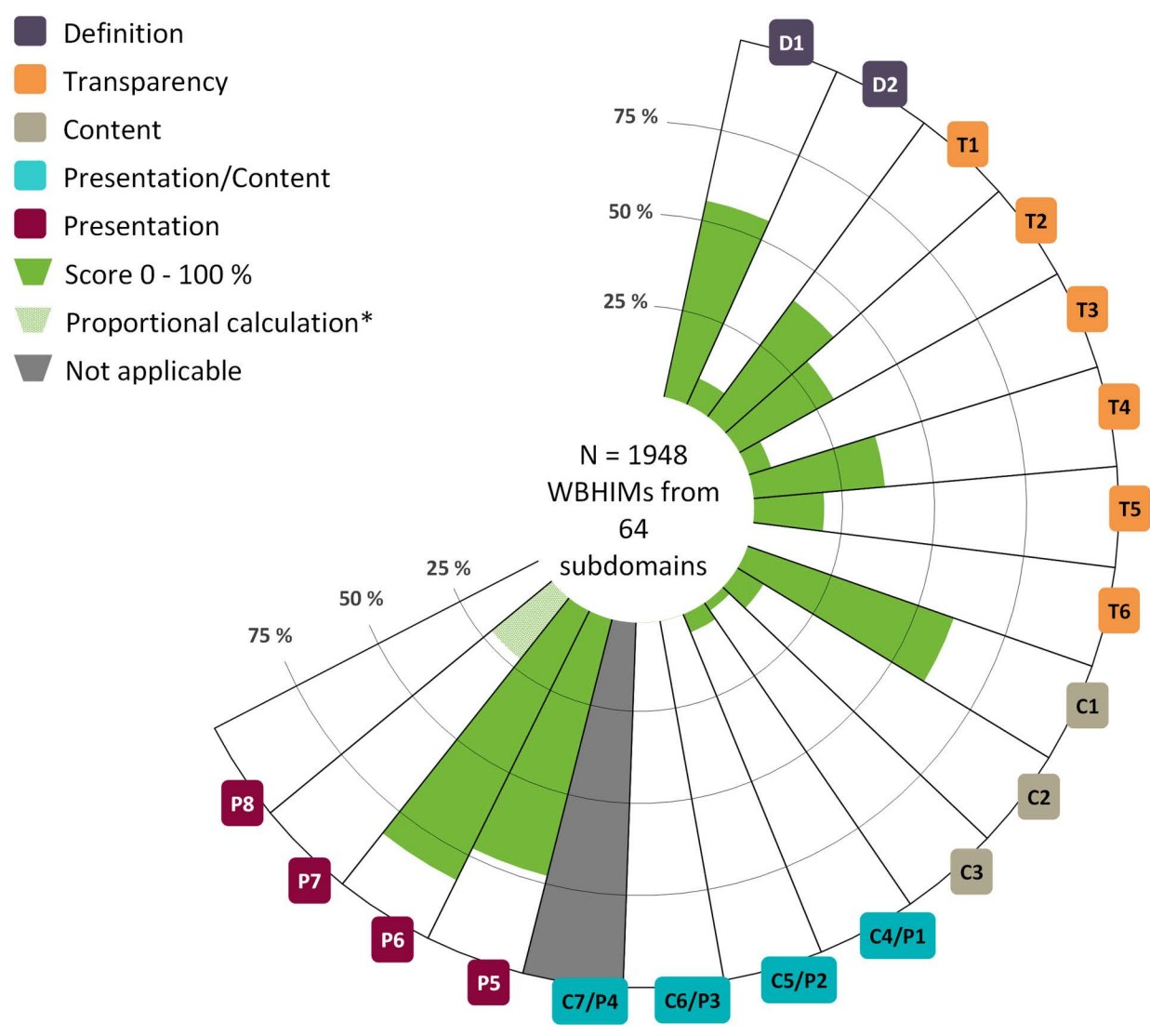

*P7: This item only applies to 30 out of 1948 health informations and was therefore calculated proportionally.

**Fig 2. Visualisation of information quality as measured with the MAPPinfo-checklist.** The figure visualises the spectrum of quality criteria of the MAPPinfo checklist. The shape of the open circle diagram indicates the nature of the underpinning concept of quality which is not complete (further research will lead to completion of the circumplex model). The green colouring gives a visual analogue impression of the extent of compliance with the respective criteria. White colouring indicates noncompliance.

Using 64 cross-sectional studies, we evaluated 1,948 WBHIMs across 16 public health domains relevant for young people from birth to legal age. The health problems included in this study can generally be managed by the citizen without involving health care providers. The MAPPinfo checklist, which operationalises current ethical and evidence-based standards for EBHI [15], was used as the evaluation method.

On average, the health information investigated meet only 22% of the criteria constituting the minimal standard. No information item approaches full compliance. Crucial content, such as the prevalence of a health problem or the potential

**Table 2. Information quality according to the 19 MAPPinfo criteria.**

| Category | | Criterion | % met (absolute number) | Std. Deviation |
|---|---|---|---|---|
| DEFINITIONS | D1 | Target group of HI | 55(1071) | 50 |
| | D2 | Aim to facilitate informed choice | 8(156) | 21 |
| TRANSPARENCY | T1 | Author indicated | 40(779) | 44 |
| | T2 | Funding declared | 30(584) | 46 |
| | T3 | COI considered | 7(136) | 25 |
| | T4 | Plan for update | 36(701) | 26 |
| | T5 | Use of references | 20(390) | 32 |
| | T6 | Reported syst. search | <1(4) | 4 |
| CONTENT | I1 | Health problem explained | 60(1169) | 49 |
| | I2 | Options named & explained | 9(171) | 22 |
| | I3 | Stochastic uncertainty explained | 3(58) | 17 |
| CONTENT/ PRESENTATION | I4/P1 | Natural course/ prevalence | 6(117) | 23 |
| | I5/P2 | Benefits presented adequately | <1(11) | 8 |
| | I6/P3 | Harms presented adequately | <1(10) | 7 |
| | I7/P4 | Presentation of test quality | n.a. | |
| PRESENTATION | P5 | Neutral language | 72(1403) | 45 |
| | P6 | Narratives not used | 82(1597) | 39 |
| | P7 | Graphics designed suitably | 22(429) | 39 |
| | P8 | Gain/loss framing | 1(2) | 3 |

The table provides numbers (percentages and absolute numbers) for the health information materials' compliance with each of the 19 criteria, which are assigned to categories in the MAPPinfo-checklist. Compliance is quantified as a percentage of the total number of 1948 information items. "n.a." = not applicable.

**Table 3. Information quality by provider classes.**

| Provider class | N of WBHIMs | QS in % | Std. Deviation | Min/max |
|---|---|---|---|---|
| scientific | 103 | 28 | 7 | 13/50 |
| NGO | 256 | 23 | 8 | 05/57 |
| governmental | 184 | 23 | 6 | 05/40 |
| commercial | 876 | 22 | 9 | 00/73 |
| health services | 203 | 21 | 6 | 05/68 |
| news | 259 | 19 | 8 | 03/50 |
| bloggers/ influencers | 67 | 19 | 7 | 05/38 |
| Total | 1948 | 22 | 9 | 00/73 |

The table provides numbers indicating the distribution of WBHIMs (=Web-based health information materials) between seven provider classes and quality scores (expressed as %) as absolute mean values for provider classes, standard deviations and the empirical range of quality scores.

benefits and harms of available treatment alternatives, is often missing. These findings raise doubts that Norwegian citizens can readily access quality information to make evidence-informed health choices.

Information quality varies only slightly between target groups and types of health problems, and even less between provider classes. Given the very small effect sizes, any statistically significant differences are likely due to the large sample size. Overall, the results indicate that poor quality of information is ubiquitous.

## Limitations

Despite combining a large number of individual surveys and screening websites, we examined only a small segment of the entire information landscape. Therefore, the generalisability of these findings should be interpreted with caution. Mapping of information quality in other health domains and languages is ongoing and will further complete the picture.

It could be argued that evaluating information used by the target groups should have included social media. Indeed, social media is increasingly used as a source of health information [17], and even governments are relying more on social media to ensure that online information is up to date and accessible [17]. However, Google – such as other search engines and large language model AI tools like ChatGPT – remains highly relevant for active information seeking. Furthermore, given the general lack of essential meta-information in social media, such as authorship, funding, source, conflicts of interest, and publication date, our results may overestimate, rather than underestimate, the quality of information across a broader range of sources, including social media.

Selecting WBHIMs using our chosen method—stepwise development and conduct of Google searches, followed by screening of references—may have introduced some degree of selection bias of uncertain magnitude. However, the risk is likely low, because the samples were large and close to capturing the respective population of information on each topic. In addition, multiple methodological safeguards were employed to minimise subjectivity, including the use of Google's incognito mode and consensus discussions within the research group regarding website inclusion. The recruitment strategy was designed to reflect the likely search behaviour of non-professional users. Any missed relevant websites were likely due to limited visibility or accessibility.

It could be argued that our evaluation addresses presentation quality only, as we did not attempt to verify the accuracy of content, such as numerical information about benefits and harms provided on the websites. By omitting an evaluation of accuracy, high-quality information might have been undervalued due to presentation weaknesses. This argument implies that it is principally possible to determine whether information is "correct," suggesting the existence of an authority that defines general truth—a position reflecting an outdated paradigm. Evidence-based practice, however, recognises that multiple answers may exist for a health question depending on the methods used to evaluate the available evidence. It is now generally accepted that no such single "truth" exists for a specific topic; instead, information quality is defined by its transparency and trustworthiness. Trustworthiness is best evaluated through the scientific methods, reasoning, and conclusions involved in developing the material [27,34,35]. Therefore, MAPPinfo does not guide the evaluator to verify the accuracy of any content. Rather, the checklist is designed to assess the material's trustworthiness and to enable a valid judgement of the quality of both content and presentation [27].

The set of quality criteria employed in this study is not exhaustive of all possible evaluative criteria. In addition to the 19 criteria of MAPPinfo and the EBHI guideline, an unknown number of other potentially relevant criteria exist. However, these criteria have yet to be identified and articulated. The concept of health information quality remains an evolving scientific field. New criteria may need to be added, and existing criteria adjusted, as the EBHI guidelines themselves are updated.

It might be argued that evaluating websites against a particular concept of quality, which providers or developers may not have intended to follow, is inappropriate. It could also be suggested that focusing on websites dedicated to facilitate informed choices may have yielded less negative results. Indeed, our sample included a high percentage of commercial websites and information from providers not primarily committed to evidence-based health choices. However, the focus of this study was to explore the quality of information to which Norwegian citizens are exposed to in reality. Thus, we did not select only those sites aiming specifically to provide high quality information. However, none of the sources evaluated met the minimum quality standards. This applied to all provider types, contradicting our hypothesis of a higher quality of information delivered from the health authorities and health care services.

## Results in the context of the literature

First, our findings need to be discussed in the context of previous research on the quality of health information. Insufficient quality has been documented in hundreds of studies across multiple countries and languages, regardless of the medical domain. Most of these studies focused on readability and understandability [36], fewer examined credibility and accuracy [37,38], and some applied evidence-based criteria [39,40]. We did not identify any studies evaluating health information using the quality criteria recommended in the EBHI guideline [15]. A recent study of officially recognised disease awareness campaigns, such as "Breast Cancer Awareness Month" or "Movember" (for men's health), found that respective websites highlighted benefits over harms of health interventions and rarely addressed potential problems such as overdiagnosis or overtreatment. These websites tended to adopt an unbalanced approach, encouraging readers to make certain choices rather than neutrally presenting available tests and treatments [41]. The current study mapped the quality of a broad range of public health information topics that readers could use independently to make their own choices independent of any health care professional. Further mapping is ongoing in other domains. However, given the ubiquitous nature of our findings, we do not anticipate significant differences in information quality across additional health domains.

A look into the specific genre of patient decision aids (PDAs), which is a subset of health information provision, demonstrates that providing high-quality information is feasible [22,23,42]. PDAs have been frequently evaluated, and some results have been impressive [23,42]. However, a rigorous focus on evidence-based criteria has not been applied systematically in all studies [24].

The Norwegian Directorate of Health adopted the International Patient Decision Aid Standards (IPDAS) [43] criteria in 2018 as a quality standard for PDAs published on the Norwegian health platform *helsenorge.no* [44]. These criteria do not fully operationalise all elements and include some that are not yet evidence-based [24]; however, the IPDAS framework largely overlaps with the MAPPinfo checklist. Nevertheless, no studies have evaluated whether Norwegian information materials comply with any of these criteria.

Our results also need to be discussed in the context of health literacy, a concept currently receiving increased attention in many countries and health systems [11–13,45–47]. For example, the Norwegian Directorate of Health has made improving citizens' health literacy a top priority [13]. This followed comprehensive surveys in Norway and other countries indicating worryingly low levels of health literacy among citizens [12]. Health literacy is defined as the ability to find, process, and use health information [11]. However, how can health literacy be adequately assessed in the absence of appropriate information? Our findings suggest that assessing health literacy without robust, high-quality health information is not possible. Consequently, published conclusions regarding population-level health literacy may be misleading. This implies that a national strategy to strengthen users' health literacy [13] needs to consider appropriate health information too.

Our results on the quality of health information accessed by the lay public should also be discussed in the context of information used by healthcare providers. Depending on the case and type of health problem, healthcare providers must educate themselves through other sources to compensate for the poor quality of information available online. This is particularly relevant in specialist medical care, where healthcare providers are directly accountable as decision makers alongside the patient.

The health problems examined in the current study can be handled independently by the lay public. However, according to professional guidelines, Norwegian PHNs are responsible for educating users in these health domains [49], suggesting that PHNs are accountable for compensating for insufficient information quality by supplementing and correcting information when the user cannot manage alone. When providing consultations, PHNs, like other health professionals, are strongly encouraged to base their advice on relevant medical guidelines. But are these guidelines always reliable sources of high-quality information? In the case of Norwegian guidelines, this does not seem to be the case. A recent systematic review of Norwegian scientific medical guidelines investigated their compliance with international standards for trustworthy clinical practice [49]. The authors used the 15 criteria of the National Evidence-based Assessment of Trustworthy Standards (NEATS) instrument [33], which are similar to the MAPPinfo criteria. Most NEATS criteria scored very

low on average across all Norwegian guidelines. The lowest scores were given to "study selection," "description of studies and results," "grading of effects reported," and "external review." The highest scores were given to "providing unequivocal recommendations," which, in the absence of transparency regarding the justification for such recommendations, might constitute another barrier to empowering the public to make informed choices.

In summary, the information sources that healthcare providers rely on are not necessarily more trustworthy than the health information available online to the general public. Consequently, improving the health literacy of healthcare providers should also be a priority for the Norwegian Directorate of Health [48].

### Implications for practice

The low quality of health information available to citizens does not appear to be a superficial problem that could be corrected simply by asking information providers to consider a few additional quality standards. Information developers might require further training to understand the published guidance and integrate this knowledge into their development processes. The outcomes of such training were evaluated in a recent German randomised study [50]; however, it did not lead to the development of higher-quality health information [51]. Considering the evidence on the quality of medical guidelines in Norway [49], addressing this problem might require more fundamental changes. To provide a reliable basis for health information development, rethinking the procedures and management of evidence updates in guideline development appears necessary, at least in the Norwegian context. Ideally, the development of medical guidelines for health professionals should automatically include the development of a corresponding guideline for the lay public. For example, the SHARE-IT project demonstrates how patient information can be automatically linked to the corresponding medical guideline [52].

As a basic requirement, agreement on the national information quality standards to be used as a template for health information development needs to be renewed or updated [44]. Currently, there is little evidence that the standards established for Norwegian PDAs are either used or fully understood by health information content developers. The scope of these quality standards should be expanded to cover health information in general, rather than being limited to PDAs.

A related challenge is determining how non-governmental providers can be motivated to adopt the same standards, given that they provide information based on various purposes (not all are aimed at facilitating informed choices). A possible solution is the implementation of a certification system, which would allow all types of health information providers to learn about and adhere to minimal quality standards via a formal code of conduct.

### Conclusions

Independent of the information provider, Norwegian health information is not of sufficient quality to facilitate the making of informed health choices. This conclusion is based on current evidence-based guidelines for health information and applies across a wide range of public health domains relevant to infants, children, and youth. Further research is needed to investigate the quality of information in other health domains and in other countries. The identified deficiencies are fundamental, affecting all facets of quality, particularly content and presentation. In the absence of health information of acceptable quality, assessing public health literacy might be unreliable, and ongoing efforts to strengthen the health literacy of Norwegian citizens may be premature.

### Supporting information

**S1 File. STROBE_checklist_MAPPinfo multi-cross-sectional.**
(DOC)

**S2 File. Overview-websides 6.**
(XLSX)

## Acknowledgments

The members of the «MAPPinfo project group» were students (or external supervisors) involved in conducting master theses as part of the project "Quality of health information: the missing link in the era of health literacy": Helene Kolnes[1], Hilde Laholt (supervisor) [1], Solveig Wergeland Hansen[2], Petter Christiansen Halse[2], Frida Kristine Jansen Broen[2], Amalie Mesel[2], Lars Peder Kolås Henriksen[2], Signe Svendsrud Sørlien[2], Ingrid Thorsen[2], Hanne Nesse[2], Christine Lilledahl[2], Charlotte Myrvollen[2], Nadia Maria Adablah Teigland[2], Linn Therese Gunnerød[2], Marte Næsgaard Myhre[2], Anna Frydenlund[2], Synnøve Skattebu[2], Mari Hårstad[2], Maiken Jacobsen[2], Anna Marie Liljenvall[2], Maren Køhn[2], and Marte Kristiansen[2]. 1. The arctic University of Tromsø, UiT, Institute of Healthcare and Health Promotion, Faculty of Health Sciences. Tromsø, Norway. 2. Oslo Metropolitan University, OsloMet, Institute of Nursing and Health Promotion, Faculty of Health Sciences, Oslo, Norway. This group is led by the first and corresponding author. We are very grateful for professional and thoughtful language editing by Dr. Wendy Longley.

## Author contributions

**Conceptualization:** Jürgen Kasper, Marianne Molin, Anke Steckelberg, Victoria T. Hjellset.

**Data curation:** Jürgen Kasper, Sandro Zacher.

**Formal analysis:** Jürgen Kasper, Sandro Zacher.

**Funding acquisition:** Jürgen Kasper, Marianne Molin, Anke Steckelberg, Victoria T. Hjellset.

**Investigation:** Jürgen Kasper, Victoria T. Hjellset, Betül Cokluk, MAPPinfo research group.

**Methodology:** Jürgen Kasper, Anke Steckelberg, Sandro Zacher, Victoria T. Hjellset.

**Project administration:** Jürgen Kasper, Betül Cokluk, Victoria T. Hjellset.

**Resources:** Jürgen Kasper.

**Software:** Sandro Zacher.

**Supervision:** Jürgen Kasper, Marianne Molin, Anke Steckelberg, Victoria T. Hjellset.

**Validation:** Jürgen Kasper, Anke Steckelberg.

**Visualization:** Jürgen Kasper.

**Writing – original draft:** Jürgen Kasper, Victoria T. Hjellset.

**Writing – review & editing:** Jürgen Kasper, Betül Cokluk, Marianne Molin, Anke Steckelberg, Sandro Zacher, Victoria T. Hjellset, MAPPinfo research group, the MAPPinfo project group.

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
