## [Decision Letter · Decision Letter 0]

16 Jul 2025

PONE-D-24-48366Mapping the quality of Norwegian health information – is it sufficiently facilitating informed choices?PLOS ONE?

Dear Dr. Kasper,

We look forward to receiving your revised manuscript.

Kind regards,

Daniel Parkes, PhD

Staff Editor

PLOS ONE

Journal Requirements:

“The study was conducted as part of the project “Quality of health information: the missing link in the era of health literacy” which is funded as a three years PHD scholarship by the faculty of health sciences, OsloMet, Metropolitan University, Oslo, Norway (Project number: 203322)”

“The study was conducted as part of the project “Quality of health information: the missing link in the era of health literacy” which is funded as a three years PHD scholarship by the faculty of health sciences, OsloMet, Metropolitan University, Oslo, Norway (Project number: 203322). The author, Betül Cokluk, is the PhD funded by tis project.”

3. One of the noted authors is a group or consortium [the MAPPinfo project group]. In addition to naming the author group, please list the individual authors and affiliations within this group in the acknowledgments section of your manuscript. Please also indicate clearly a lead author for this group along with a contact email address.

Reviewers' comments:

Reviewer's Responses to Questions

**Comments to the Author**

1. Is the manuscript technically sound, and do the data support the conclusions?

Reviewer #1: Partly

Reviewer #2: Yes

2. Has the statistical analysis been performed appropriately and rigorously?

Reviewer #1: I Don't Know

Reviewer #2: Yes

3. Have the authors made all data underlying the findings in their manuscript fully available?

Reviewer #1: Yes

Reviewer #2: No

4. Is the manuscript presented in an intelligible fashion and written in standard English?

Reviewer #1: No

Reviewer #2: Yes

Reviewer #1: Thank you for the opportunity to review this manuscript. Congratulations on the effort required to bring this work to the point of submitting for publication.

This is a cross-sectional survey of health information available on the Internet in Norway. The authors seek to map the quality of web based health information across 64 domains. I will provide feedback by section.

Background

This section has quite a few good ideas but the language contained is very difficult to parse. Some claims do not make sense given the awkward language – see Line 61-62 “External authorities such as doctors, health authorities or insurances are no longer capable of approving decisions.” The logic here as written doesn’t make sense – physicians do approve treatment decisions, health authorities control treatments through programming, and insurance companies will not reimburse for treatments it refuses to cover. I think what the authors are saying is that individuals are partners in their treatment with their physicians, with health authorities and insurance companies having a role in determine what is available but not determining an individual’s choice. Unfortunately, this section has multiple examples of where the language makes it very difficult to understand what the author is saying.

There are numerous examples of awkward language throughout the section. For example, Line 37 in the Abstract: “Our study aimed at mapping…” instead of “Our study aimed to map…”. At Line 60 “no one other than the person concerned has the sovereignty…” – an English speaker would not typically describe the concept of personal autonomy as sovereignty, a concept more to do with nations than people. Colloquialisms such as ‘as matter of fact’ (Line 77) are best left outside the text. Word choice should be reconsidered. One example is on Line 85 where ‘competences’ should read as ‘competencies’. The concepts that the authors write about all seem very reasonable. The language, however, reads extremely awkwardly and is not suitable for publication in an English-language journal in its current form.

Methods

It is difficult for this reviewer to ascertain the nature of the design of the study. The scope appears enormous (Line 138: “The current article merges 64 cross-sectional studies in 16 health domains) but I cannot tell from the manuscript design section what the design was and how it was executed. I don’t see a figure that describes the design or accompanying text. In the section labelled ‘Sample’, the authors describe a focus on infant health, then child health, then youth health. These concepts are introduced without prior discussion and it’s not clear why these populations were selected given that the introduction section made no mention of anything to do with children or youth. The section continues to describe steps in what appear to be subject engagement who collected data that the author’s used. This is particularly confusing given that there is no ethics approval despite the fact human subjects appear to be involved. The authors cite the Norwegian agency for shared services in education and research but it’s not clear that this is an ethics board approval acceptable for research publication. The authors state in the disclosure that ethics does not apply here, but any human subjects research requires ethics approval.

The statistical methods are described, but with limited detail. I suggest this be improved to describe what each statistical test is attempting to do, as some of the measures (T-coefficients, post-hoc Scheffe tests, etc.) are likely to be unfamiliar to many readers. Significant improvement here is necessary.

Despite having read this section several times, I’m still not sure exactly what the steps were in the data collection process. A figure is badly needed here to explain exactly what is going on. The authors need to redevelop this section.

Results

Table 1 is logically presented and has useful information to the reader. However, additional text is required to describe its contents and deserves several sentences identifying key findings relevant to the table. It should not be combined with a description of Figure 1.

Figure 1 attempts to communicate something but I cannot determine what. The figure itself is not described in the text as a unique callout, something that requires remedying. I suggest that the authors pull out Figure 1 and Table 1 and write text that explains to the reader what they are seeing. As it stands, it’s very difficult to understand.

Table 2 receives very little descriptive text. Considerably more discussion of the results contained therein are necessary.

Statistical results from the ANOVA are presented without reference to results in a table. I would have expected to see some tabular reporting. There is a lot of notation in this section (Lines 329 to 339) that I struggle to make sense of. Table 3 is referenced on Line 342 but I can’t match what is in the text with what is in the table.

Discussion & Conclusion

The authors provide reasonable discussion. Results are contextualized and limitations are reported. The conclusion does appear aligned to the methods and approach. The language issues noted above continue throughout. The English writing is a severe limitation and is taking away from what otherwise, could be an interesting and relevant manuscript to the people of Norway, and one that warrants consideration for publication. However, in its current form, I cannot understand from the manuscript what was done, why it was done, and how it was done, such that I can confidently judge the discussion of the findings and conclusions drawn by the authors.

Overall Comments

This is an interesting concept – mapping the quality of health data on the Internet – and one that needs to be studied in this age of misinformation. The authors clearly have done a great deal of work to analyze hundreds of data sources and critically appraise them. Unfortunately, that effort is not reflected in the quality of the manuscript. The use of English is very difficult to understand and requires significant improvement to be suitable for publiaction. The description of the methods is vague and difficult to follow. To wit, I could not replicate the author’s methods from the text provided. I can make out clearly that there is a lot of thought put in, but I don’t see that reflected in the text in a way that I can understand it. Similarly, statistical methods are inadequately described. Results are also difficult to understand, though again, there is evidence of the author’s thought, it just isn’t reflected adequately in text.

My largest concern is about the lack of ethics approval. There is a clear statement of the use of human subjects to gather information to inform the study design, yet there is no ethics disclosure, and the approval cited appears to be an organizational approval and not an ethics approval. The authors must clarify this. If there is no ethics approval needed, these details need to be clarified in the context of the information provided in the methods section. The incongruence between the disclosure at the front of the paper and the text of the article is jarring and needs to be accounted for.

Significant further work is necessary to ready this manuscript for publication.

Reviewer #2: The authors have written an interesting paper. Some of the english language is not written so academically and could use a revision. Example: line 471: which - by the way - are quite similar to the MAPPinfo criteria. The authors have reviewed websites as health information source while todays generation relies on sources on social media other than websites, it would be interesting to know why the authors focussed on websites only. The authors refer to 64 cross-sectional studies on several places, but this is unclear. How were these studies identified? In other places it is mentioned 64 populations or 64 subdomains.

Some minor comments:

- Line 109: add youth or adolesents where it is written children, as age is up to 20

- Remove all bolt words in the discussion

- Line 364, very unclear sentence, rephrase.

- It should be explained why ´content´ was not evaluated of the information reviewed.

- Line 396 - Evidence-based practice, however, taught us to accept the possibility of different good answers to the same question depending on the methods used in the evidence update. Having in mind the lack of a generally accepted truth, quality is today defined referring to transparency and trustworthiness of a development. - Rather vague sentence. Could be removed?

Line 450: But how can health literacy be assessed in the absence of

appropriate information? Our findings imply that basic requirements for measuring health literacy of

the public are not met, - but the current study is not accessing health literacy - so unclear why this is relevant.

**Do you want your identity to be public for this peer review?** For information about this choice, including consent withdrawal, please see our Privacy Policy

Reviewer #1: No

Reviewer #2: No

---

## [Author Response · Author response to Decision Letter 1]

29 Aug 2025

All Responses are included in the letter, (Kasper 20 07 25 03-WLEdits190825 R1 Letter) uploaded on this platform.

---

## [Decision Letter · Decision Letter 1]

17 Nov 2025

PONE-D-24-48366R1Mapping the quality of Norwegian health information –Does it facilitate informed choices?PLOS ONE?

Dear Dr. Kasper,

Thank you for submitting your manuscript to PLOS ONE. After careful consideration, we feel that it has merit but does not fully meet PLOS ONE’s publication criteria as it currently stands. Therefore, we invite you to submit a revised version of the manuscript that addresses the points raised during the review process.

We look forward to receiving your revised manuscript.

Kind regards,

Helen Howard

Staff Editor

PLOS ONE

Journal Requirements:

Reviewers' comments:

Reviewer's Responses to Questions

**Comments to the Author**

Reviewer #1: (No Response)

Reviewer #2: All comments have been addressed

2. Is the manuscript technically sound, and do the data support the conclusions?

Reviewer #1: Partly

Reviewer #2: Yes

3. Has the statistical analysis been performed appropriately and rigorously?

Reviewer #1: Yes

Reviewer #2: Yes

4. Have the authors made all data underlying the findings in their manuscript fully available?

Reviewer #1: Yes

Reviewer #2: Yes

5. Is the manuscript presented in an intelligible fashion and written in standard English?

Reviewer #1: No

Reviewer #2: No

Reviewer #1: The manuscript is much improved with the concerns largely addressed. However, the manuscript is still not ready for publication. Language, while improved, still is not to academic standard. Passive voice is used throughout (e.g., "The quality of the measurement" (Line 341), "The quality of the information" (Line 349, etc. Some language is awkward (e.g., does it matter that Master's students were used? I don't think so, they're research assistants in my mind). Titles for figures also contain awkward phrasing. (e.g., "Fig 1: Title: Overview over the procedure of the multi-cross-sectional study"The tables are not publication quality and do not use standard formats for things like table footnotes (i.e., asterisk, daggers, etc.). While the STROBE statement is followed, the use of headings is not consistent. The paper refers to chapters, which is not correct.

On the whole, this is a much improved draft but it is still not ready for publication in an English-language journal. The major issues that remain are formatting and language, especially as PLOS ONE does not copy edit manuscripts.

This said, the work is interesting and the findings are reasonable given the methods. Certainly, to find out that most health information doesn't meet minimum standards is a worthwhile finding deserving of publication. A further reworking of the text should get the job done.

Reviewer #2: Unfortunately, I was unable to find the letter with point-by-point comments to the questions raised. However, while reviewing the edited manuscript I see much has been changed by the authors and especially the methodology is more clear. The manuscript is dense and lengthy, some of the additional text provided helps to offer clarity in some sections, but the text could benefit from copy editing. Some of the sentences have grammatical errors or are difficult to read.

**Do you want your identity to be public for this peer review?** For information about this choice, including consent withdrawal, please see our Privacy Policy

Reviewer #1: No

Reviewer #2: **Yes:** Andrea Solnes Miltenburg

---

## [Author Response · Author response to Decision Letter 2]

4 Dec 2025

We are grateful for both reviewers' comments. Our reaction was to employ a professional language editor (Wiley) to do a thorough evaluation of the English. This was very helpful. We also did some slight changes to the formatting of the tables.

---

## [Decision Letter · Decision Letter 2]

7 Jan 2026

PONE-D-24-48366R2Mapping the quality of Norwegian health information –Does it facilitate informed choices?PLOS One?

Dear Dr. Kasper,

Thank you for submitting your manuscript to PLOS ONE. After careful consideration, we feel that it has merit but does not fully meet PLOS ONE’s publication criteria as it currently stands. Therefore, we invite you to submit a revised version of the manuscript that addresses the points raised during the review process.

We look forward to receiving your revised manuscript.

Kind regards,

Jianhong Zhou

Staff Editor

PLOS One

Journal Requirements:

Reviewers' comments:

Reviewer's Responses to Questions

**Comments to the Author**

Reviewer #1: (No Response)

Reviewer #2: All comments have been addressed

2. Is the manuscript technically sound, and do the data support the conclusions?

Reviewer #1: Yes

Reviewer #2: Yes

3. Has the statistical analysis been performed appropriately and rigorously?

Reviewer #1: (No Response)

Reviewer #2: Yes

4. Have the authors made all data underlying the findings in their manuscript fully available?

Reviewer #1: Yes

Reviewer #2: Yes

5. Is the manuscript presented in an intelligible fashion and written in standard English?

Reviewer #1: Yes

Reviewer #2: Yes

Reviewer #1: Thank you for the opportunity to review this final manuscript revision. Congratulations on the effort required to revise this work. I appreciate the efforts of the authors to incorporate the suggestions of the peer reviewers.

My concerns about the last version were rooted in readability – these have largely been addressed. I note a few minor issues that I’ll mention here to help the authors finalize their manuscript before final consideration is given by the PLOS editorial group.

Line 61…delete “and permitted” as I think this word choice introduces a bunch of ideas that I don’t think are needed here (i.e., patient autonomy) that I think would warrant more discussion to fully address. Suggest leaving it out.

Line 64…replace “enable” with “enabling” for correct verb tense.

Line 68…delete “the” in front of “informed choice”, it’s not necessary.

Line 75…suggest deleting ‘, or even most,’ as it is distracting and doesn’t add anything.

Line 108…delete ‘even’, it’s a bit conversational and not needed.

Line 111…consider capitalizing Public Health Nurses before the acronym and drop the ‘s’ from (PHNs) as it’s not needed in the bracketed acronym.

Line 122…might want to soften this sentence to say “To the best of our knowledge, this type of research has not been conducted in Norway….”

Line 146…the sentence beginning with “Focussing on the reality….” needs revision as it reads awkwardly. Suggest “We designed this study from the user’s perspective given that it is the individual who seeks out health information in the majority of cases and thus to whom we want to generalize our results.” or something to that effect. It’s the right idea, just needs a bit of revising.

Line 205 – 209…the authors may wish to consider alternative formatting…numbering the items in text might be better than calling them out as a numbered list.

Line 251…delete (MS) as it’s not relevant, Excel is not all capitalized and include the version number that was used. Software citations do follow a format, usually along the lines of ‘Ratings were documented in Excel Version 2504 (Microsoft Corporation)…”

Line 318…this probably belongs in a footnote or elsewhere in text; it’s kind of hanging out on its own and it’s not clear what the data file is.

Line 344…the research questions should be introduced in the Introduction. They read awkwardly here. Revision is necessary.

Line 370….strip out the bracketed text in the heading, the title label should speak for itself.

Line 425…extra space between ‘the’ and ‘target’ needs to be deleted.

Line 428…reference to Google should probably be broader to something that includes search engines and indeed, large language model AI tools like ChatGPT.

Line 469…this section needs to be before limitations.

Line 555…Suggest replacing “Regardless” with “Independent” as it has a bit more of a scientific flavour to it.

Overall, this is a much-improved manuscript. I think these suggestions are largely cosmetic at this point. Congratulations again to the authors on piece of work that answers a really relevant question about health information available online.

Reviewer #2: The current version of the article is much better. I have no further comments that need to be addressed.

**Do you want your identity to be public for this peer review?** For information about this choice, including consent withdrawal, please see our Privacy Policy

Reviewer #1: No

Reviewer #2: **Yes:** Andrea Solnes Miltenburg

---

## [Author Response · Author response to Decision Letter 3]

8 Jan 2026

Dear Editors, a revision letter and a point by point response are uploaded as files.

Best regards

Jürgen Kasper

---

## [Editor Report · Decision Letter 3]

29 Jan 2026

Mapping the quality of Norwegian health information –Does it facilitate informed choices?

PONE-D-24-48366R3

Dear Dr. Kasper,

We’re pleased to inform you that your manuscript has been judged scientifically suitable for publication and will be formally accepted for publication once it meets all outstanding technical requirements.

Kind regards,

Jianhong Zhou

Staff Editor

PLOS One
---

## [Editor Report · Acceptance letter]

PONE-D-24-48366R3

PLOS One

Dear Dr. Kasper,

I'm pleased to inform you that your manuscript has been deemed suitable for publication in PLOS One. Congratulations! Your manuscript is now being handed over to our production team.

Kind regards,

on behalf of

Dr. Jianhong Zhou

Staff Editor

PLOS One